# Effect of Absorbent Foam Filling on Mechanical Behaviors of 3D-Printed Honeycombs

**DOI:** 10.3390/polym12092059

**Published:** 2020-09-10

**Authors:** Leilei Yan, Keyu Zhu, Yunwei Zhang, Chun Zhang, Xitao Zheng

**Affiliations:** 1School of Aeronautics, Northwestern Polytechnical University, Xi’an 710072, China; yanleilei@nwpu.edu.cn (L.Y.); zhukeyu@mail.nwpu.edu.cn (K.Z.); c.zhang@nwpu.edu.cn (C.Z.); 2Aeronautics Engineering College, Air Force Engineering University, Xi’an 710051, China; zhang_yunwei@126.com

**Keywords:** absorbent polymethacrylimide foam, honeycomb, electromagnetic wave absorption, compressive behavior

## Abstract

Polylactic acid (PLA) hexagonal honeycomb structures were fabricated by using 3D-printing technology. By filling with absorbent polymethacrylimide (PMI) foam, a novel absorbent-foam-filled 3D-printed honeycomb was obtained. The in-plane (L- and W-direction) and out-of-plane (T-direction) compressive performances were studied experimentally and numerically. Due to absorbent PMI foam filling, the elastic modulus, compressive strength, energy absorption per unit volume, and energy absorption per unit mass of absorbent-foam-filled honeycomb under L-direction were increased by 296.34%, 168.75%, 505.57%, and 244.22%, respectively. Moreover, the elastic modulus, compressive strength, energy absorption per unit volume, and energy absorption per unit mass, under W-direction, also have increments of 211.65%, 179.85, 799.45%, and 413.02%, respectively. However, for out-of-plane compression, the compressive strength and energy absorption per unit volume were enhanced, but the density has also been increased; thus, it is not competitive in energy absorption per unit mass. Failure mechanism and dimension effects of absorbent-foam-filled honeycomb were also considered. The approach of absorbent foam filling made the 3D-printed honeycomb structure more competitive in electromagnetic wave stealth applications, while acting simultaneously as load-carrying structures.

## 1. Introduction

Multifunctional design of lightweight hybrid structure can make it competitive in aircraft and aerospace applications, such as combination of load carrying and electromagnetic wave absorption. Lattice structures (such as corrugated cores [1] and honeycomb cores [2,3]) have advantages in load bearing [4] and impact energy-absorption applications [5,6,7], due to their high specific strength and specific energy absorption (SEA). In addition, honeycombs by foam filling or resistive-films attachment [8,9,10,11] can also have advantages in electromagnetic wave absorption. Jiang et al. [8] designed a honeycomb structure exhibiting broadband absorbing performance with an ultra-thin thickness through 3D printing and silk-screen printing. In this paper, the mechanical properties are improved based on the honeycomb structure, to meet the combination of load-carrying and electromagnetic-wave absorption.

Recently, some novel honeycomb-based structures were proposed, such as embedded [12,13], tandem [14,15,16], hierarchical [17], and negative Poisson’s Ratio (NPR) [18] honeycombs. Moreover, filling tube [19,20,21,22] and foam [23,24,25,26,27,28] into honeycomb was also regarded as an efficiency method to increase the mechanical and energy-absorption performances. Hussein et al. [20] studied the axial crushing response of aluminum honeycomb-filled square carbon-fiber-reinforced plastic (CFRP) tubes, showing increased mean crushing force and energy absorption. Zhang et al. [21] proposed a novel metallic tube-reinforced honeycomb, and its quasi-static compression and three-point bending were investigated experimentally and numerically. The method of tube filling shows extensive potential in improving mechanical properties. Moreover, foam, as a lightweight material, was also used as a filling material, to increase the mechanical properties of honeycomb, due to the lateral support supplied by foam which stabilized their core members. Liu et al. [24] investigated the compressive response of empty honeycomb and honeycomb filled with Expanded Polypropylene (EPP) foam. The mechanical properties and energy-absorption characteristics had been greatly improved, especially under lateral crushing. Moreover, polyurethane foam [25,28] with great mechanical performance, carbon foam [29], and graphene [30], exhibiting a great electromagnetic absorption capacity, make the multifunctional design more selective.

Unlike the conventional fabrication approaches, 3D printing has become a new manufacturing method for honeycomb [31,32,33,34,35,36,37]. With the development of 3D-printing technology, the manufacturing of complex topologies has become possible. For instance, Chen et al. [32] fabricated a hierarchical honeycomb by using 3D printing, showing a progressive failure mode under uniaxial compression, along with increased stiffness and energy absorption. Tao et al. [36] investigated mechanical properties and energy absorption of square hierarchical honeycombs (SHHs) fabricated by 3D printing. They elucidated the effect of structural hierarchy on the mechanical properties and energy-absorption performance of regular cellular materials. Thus, it was a very attractive idea to fabricate a novel honeycomb structure via 3D-printing technology, to broaden the application range of honeycomb. Compared with conventional material, internal flaws will be created during the 3D-printing fabrication process, which may cause a decrease in mechanical properties and limit the application of 3D printing in honeycomb structures.

Therefore, the purpose of present study was to design a hybrid structure with excellent mechanical performance and electromagnetic-wave-absorption properties, based on previous study [8,38]. The absorbent polymethacrylimide (PMI) foam with absorptivity exceeding 85% at a large frequency range of 4.9–18 GHz and 3D-printed honeycomb with absorptivity exceeding 90% at a large frequency range of 3.53–24.00 GHz were offered. By filling with absorbent PMI foam, a novel absorbent foam filled honeycomb was obtained. The mechanical behaviors of in-plane (L- and W-direction) and out-of-plane (T-direction) compression were studied experimentally and numerically. The typical stress–strain curves and failure mechanisms of empty and foam-filled honeycomb were discussed and compared. Moreover, the filling effects and enhancement of mechanical properties were also discussed in detail.

## 2. Materials and Methods

### 2.1. Description of Specimens

The design and geometry parameters of empty and absorbent foam-filled honeycomb are illustrated in Figure 1. By carbon-based electromagnetic-absorbing-agent-filling during the foaming process, absorbent PMI foam (materials supplied by Hunan Zihard Material Technology Co., Ltd., Hunan, China) was obtained that exhibited an excellent electromagnetic-wave-absorbing property. The hexagonal honeycomb was fabricated by using a material of polylactic acid (PLA) (Yisheng Co., Ltd., Shenzhen, China), which was biodegradable and bioactive thermoplastic aliphatic polyester, and 3D-printing technology (Fused deposition modeling, FDM) was adopted for the preparation process. The PLA material was heated and melted, to become semi-liquid. Then the prepared PLA material was squeezed out through the nozzle on a predetermined route, and stacked layer by layer. The 3D-printed PLA honeycomb was obtained after cooling and curing. The density of absorbent PMI foam and PLA material was 222 and 1240 kg/m^3^, respectively. Then the absorbent PMI foam was cut by an electronic automatic-cutting machine, for uniformity, and filled into hexagonal honeycomb. The absorbent foam and hexagonal honeycomb were assembled and subsequently stuck together with glue, to get the absorbent-foam-filled honeycomb.

The dimensions of hexagon side length (*L*_h_ = 8 mm), cell wall thickness (*T*_h_ = 1.9 mm), and honeycomb core height (*H*_c_ = 15.51 mm) were all fixed for honeycomb unit cell. Moreover, the dimensions of length (*L*_c_ = 81.87 mm) and width (*W*_c_ = 78.78 mm) were also considered [8]. Typical specimens of empty honeycomb and absorbent-foam-filled honeycomb were shown in Figure 2. The electromagnetic-wave absorption of the absorbent PMI foam and PLA 3D-printed honeycomb based on metamaterial absorber were measured in earlier studies [8,38]. The PLA 3D-printed honeycomb with resistive films and carbon-based absorbent PMI foam exhibited excellent electromagnetic-wave-absorption properties.

### 2.2. Quasi-Static Compressive Test Process

In-plane and out-of-plane compression tests on prepared specimens were carried out by the electromechanical universal testing machine (INSTRON-8803, INSTRON, BOSTON, AMERICA), at ambient temperature. The specimen was sandwiched between two rigid cylindrical plates. In order to satisfy the quasi-static compression condition, the displacement rate was set to 4 and 0.5 mm/min, under in-plane and out-of-plane, with nominal strain rate less than 10^−3^ s^−1^, respectively. Moreover, the deformation processes of the specimens were recorded, and the load-displacement curves were also obtained.

### 2.3. Numerical Simulations

In-plane and out-of-plane compression responses of empty honeycomb and absorbent-foam-filled honeycomb were modeled by commercial finite element software (ABAQUS 2017/Explicit, SIMULIA) to predict the mechanical behaviors, along L-, W-, and T-direction. As shown in Figure 3, the specimens were sandwiched between two rigid surfaces simulating the platens of the INSTRON-8803.

The basic materials parameters of polylactic acid and absorbent PMI foam were obtained in the FE model by stress–strain curves, which were obtained by the experimental test (Figure 3b). Moreover, the density of polylactic acid and absorbent PMI foam was 1.24 and 0.222 g/cm^3^, respectively, which is similar to specimens for compressive tests. The polylactic acid and absorbent PMI foam were meshed with the C3D8R hexahedral 8-nodes linear element with reduced integration. The contact included normal and tangential behaviors based on penalty formulation, and the friction coefficient was set to be 0.3. The “General contact” was used to prevent the probable interpenetration. To ensure the simulation was quasi-static, the kinetic energy was controlled to be lower than 5% of the total energy in the system.

## 3. Results and Discussion

### 3.1. Experimental Analysis

#### 3.1.1. In-Plane Compression Responses

Quasi-static compressive tests were conducted to investigate mechanical behaviors of absorbent-foam-filled honeycomb and empty honeycomb. The compression test results of specimens under in-plane compression are summarized in Table 1, and stress–strain curves of the specimens are shown in Figure 4. For empty honeycomb under L-direction, the stress reached its compressive strength, $\sigma_{peak}$ (the first peak stress), and rapidly declined to a low value. Then the stress remained at a low value (almost equal to zero) until the strain was 0.16, and then it started to rise. After the compressive stress increased to a large value (less than the first peak stress), the compressive stress decreased rapidly again. The jagged stress–strain curves presented multiple valleys and peaks. From corresponding deformation processes in Figure 5, the honeycomb cell walls of the empty honeycomb under L-direction (L-EH-1) appeared brittle and fractured. Once a honeycomb cell wall was broken in one place, the surrounding honeycomb cell walls buckled and moved toward the broken area, finally squeezing each other and causing a decrease of the stress and the fluctuation of stress–strain curves. It indicated that each formation of folding for honeycomb cell wall was related to one stress peak and valley in Figure 4b.

Unlike empty honeycomb under L-direction (L-EH), the absorbent-foam-filled honeycomb (Figure 4a, L-FH-1) under L-direction exhibited foam-like features, i.e., the three typical regions, namely linear, plateau, and densification regions [38]. Figure 4a shows that absorbent-foam-filling led to a significant reinforcement of the compressive performance, as compared with the empty honeycomb. After a linear and nonlinear increase, the compressive stress of absorbent-foam-filled honeycomb under L-direction reached its peak strength (5.96 MPa) and subsequently decreased to a lower plateau value of 3.67 MPa. The compressive stress of foam-filled honeycomb (L-FH-1) maintained a plateau value until entering the densification stage at ε=0.46. Compared with empty honeycomb, the average elastic modulus and compressive strength of absorbent-foam-filled honeycomb were increased by 296.34% and 168.75%, respectively. From compressive deformation histories in Figure 5, due to the absorbent foam filling, the foam acted as a support and connection to the walls of the honeycomb, and the structural integrity could also be well ensured. When a cell wall of the honeycomb was broken, the foam limited the relative motion of the surrounding honeycomb cell wall, and the absorbent-foam-filled honeycomb could continue to resist deformation as a whole; this also explains why the stress–strain curve exhibited a plateau region.

As shown in Figure 4b, the compressive responses of foam-filled honeycomb and empty honeycomb under W-direction were also similar to those under L-direction. The empty honeycomb under W-direction (W-EH) showed a poor compressive performance, and the compressive response was strengthened due to the foam filling. The average elastic modulus and compressive strength of the absorbent-foam-filled honeycomb under W-direction were increased by 211.65% and 179.85%, respectively. Overall, as with the compression response under L- and W-direction discussed before, the method of foam filling was proved to be effective in increasing the in-plane compressive performance of 3D-printed honeycomb.

#### 3.1.2. Out-Of-Plane Compression Response

Nominal stress–strain curves and the corresponding deformation processes of empty honeycomb and absorbent-foam-filled honeycomb under out-of-plane (T-direction) compression were shown in Figure 6. Compared with in-plane compression, empty honeycomb and absorbent-foam-filled honeycomb under T-direction exhibited larger compressive strength. For absorbent-foam-filled honeycomb, the compressive strength increased from 14.052 to 17.350 MPa, and the corresponded strain increased from 0.051 to 0.074, compared to the empty honeycomb. Moreover, due to the foam filling, the densification stage of the honeycomb happened forward. The densification strain of absorbent-foam-filled honeycomb declined from 0.601 to 0.403, exhibiting higher densification stress. From deformation process of empty honeycomb (Figure 6b), with the increase of compressive strain, the honeycomb cell walls gradually thickened, resulting in a decrease in the space enclosed by the honeycomb cell walls. Besides, the filling of absorbent foam slowed the process of the increase of wall thickness for empty honeycomb and made honeycomb and absorbent foam squeezed each other, which explain the earlier occurrence of the densification stage.

Besides elastic modulus and compressive strength, the energy-absorption characteristic was also an important parameter to evaluate mechanical behaviors. Energy absorption of per unit volume ($W_{v}$) was commonly used to represent energy-absorption capacity, which could be obtained through the integration of the stress–strain curves:(1)$$W_{v}=\int_{0}^{\varepsilon }\sigma d\varepsilon$$
where the strain of specimens ε=0.5 was adopted here.

Moreover, the specific energy absorption (SEA) was also another important parameter for weight-sensitive applications, which could be defined as follows [39]:(2)$$W_{m}=\frac{W_{v}}{\rho_{c}}$$
where the $\rho_{c}$ was defined as the total mass of specimen divided by the whole volume (81.87 mm × 78.78 mm × 15.51 mm).

The energy absorption per unit volume and per unit mass of specimens are summarized in Table 1. The energy-absorption comparisons of both empty and absorbent-foam-filled 3D-printed honeycomb under three directions (L-, W-, and T-direction) are shown in Figure 7. The results in Table 1 and Figure 7 indicate that, due to the absorbent foam filling, the energy absorption per unit volume and per unit mass of absorbent-foam-filled honeycomb under L-direction were increased by 505.57% and 244.22%, respectively. Moreover, the energy absorption per unit volume and per unit mass of absorbent-foam-filled honeycomb under W-direction were increased by 799.45% and 413.02%, respectively. For absorbent-foam-filled honeycomb under T-direction, the energy absorption per unit volume increased by 28.59%, and the energy absorption per unit mass decreased by 26.85%. In conclusion, the method of filling with absorbent foam could increase energy-absorption characteristic greatly under in-plane compression, but it is not effective for out-of-plane compression (T-direction).

### 3.2. Finite Element Analysis

From the conclusions in Section 3, the method of filling with absorbent foam could improve the mechanical performance and energy-absorption characteristic of 3D-printed honeycomb under in-plane compression greatly, but there was almost no influence for 3D-printed honeycomb under out-of-plane compression. Therefore, to better understanding the effect of absorbent foam filling on mechanical behaviors of 3D printed honeycomb, FE simulation was carried for absorbent foam filled honeycomb with different cell wall (*T*_h_ = 1.4, 1.9, and 2.4 mm) under in-plane compression. As shown in Figure 8, the typical simulated stress–strain curve of the empty honeycomb and absorbent-foam-filled honeycomb under L-direction had a good agreement with the experimental measurement results.

The compressive performance of the absorbent-foam-filled honeycomb with different cell-wall thickness (*T*_h_ = 1.4, 1.9, and 2.4 mm) was studied by FE analysis. As shown in Figure 9, the energy absorption per unit mass of absorbent-foam-filled honeycomb increased with the increase in thickness. Compared with empty honeycomb, the energy absorption per unit mass of absorbent-foam-filled honeycomb (*T*_h_ = 1.4, 1.9, and 2.4 mm) under L-direction was increased by 306.5%, 258.62%, and 421.93%. Moreover, the energy absorption per unit mass of absorbent-foam-filled honeycomb (*T*_h_ = 1.4, 1.9, and 2.4 mm) under W-direction was increased by 410.4%, 412.6%, and 523.3%, respectively. It indicated that the absorbent-foam-filled honeycomb under in-plane compression, with greater cell-wall thickness, had a higher energy-absorption characteristic.

## 4. Conclusions

In conclusion, by filling with absorbent polymethacrylimide (PMI) foam, a novel absorbent-foam-filled honeycomb based on 3D-printing technology was obtained. Experimental and numerical approaches were employed to investigate the in-plane and out-of-plane compressive responses of absorbent-foam-filled honeycomb. The main findings are summarized as follows:

(1) During the in-plane compression process, the empty honeycomb appeared cracked and fractured, showing poor mechanical performance. In contrast, the absorbent-PMI-foam honeycomb exhibited greater compression response, due to the addition of absorbent PMI foam. The filling of absorbent foam changed the deformation mode of empty honeycomb and allowed the absorbent-foam-filled honeycomb to have excellent mechanical and electromagnetic-wave-absorption performances.

(2) The normalized elastic modulus, *E*, compressive strength, $\sigma_{peak}$, energy absorption per unit volume, $W_{v}$, and per unit mass, $W_{m}$, of absorbent-foam-filled honeycomb under L-direction were increased by 296.34%, 168.75%, 505.57%, and 244.22%, respectively, and under W-direction, they were 211.65%, 179.85, 799.45%, and 413.02%, respectively. It indicated that the approach of using the absorbent foam filling greatly strengthened the mechanical properties and energy-absorption characteristics of the empty honeycomb under in-plane compression. For absorbent-foam-filled honeycomb under out-of-plane compression, the compressive strength was increased by 23.5%, but there was almost no improvement on elastic modulus and energy-absorption characteristic.

(3) With their outstanding performances in electromagnetic-wave absorption, compressive strength, and energy absorption, the proposed absorbent-foam-filled honeycomb is quite competitive in applications such as simultaneous electromagnetic wave stealth, load carrying, and impact resistance.

## Figures and Tables

**Figure 1 polymers-12-02059-f001:**
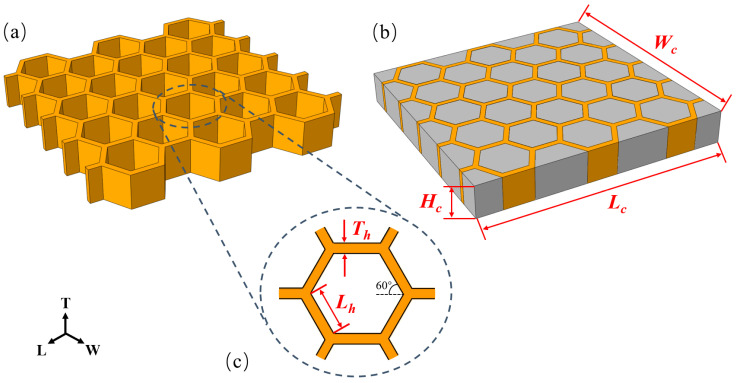
Schematic and parameter of specimens: (**a**) empty honeycomb, (**b**) absorbent-foam-filled honeycomb, and (**c**) detail of honeycomb unit cell.

**Figure 2 polymers-12-02059-f002:**
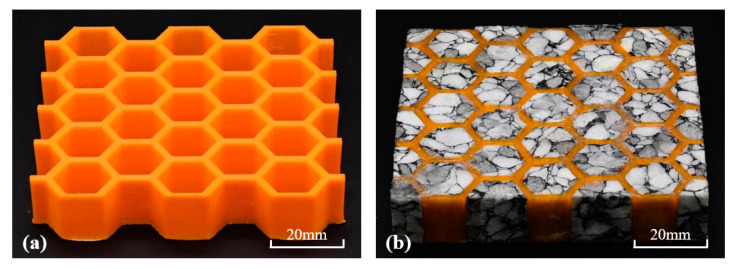
Images of empty honeycomb and absorbent-foam-filled honeycomb: (**a**) empty honeycomb and (**b**) absorbent-foam-filled honeycomb.

**Figure 3 polymers-12-02059-f003:**
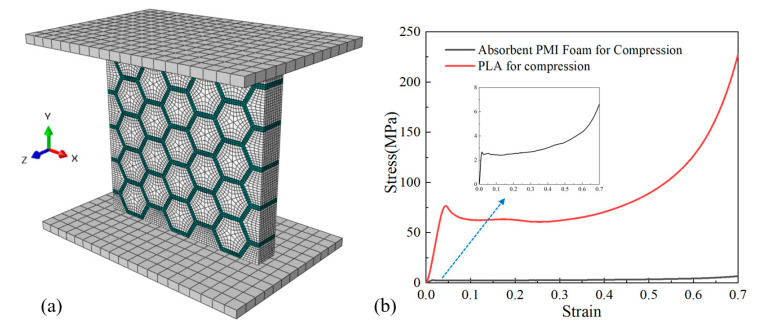
Details of numerical model for quasi-static compression and required stress–strain curves: (**a**) FE model and (**b**) measured compression stress–strain curves for absorbent polymethacrylimide (PMI) foam and polylactic acid (PLA).

**Figure 4 polymers-12-02059-f004:**
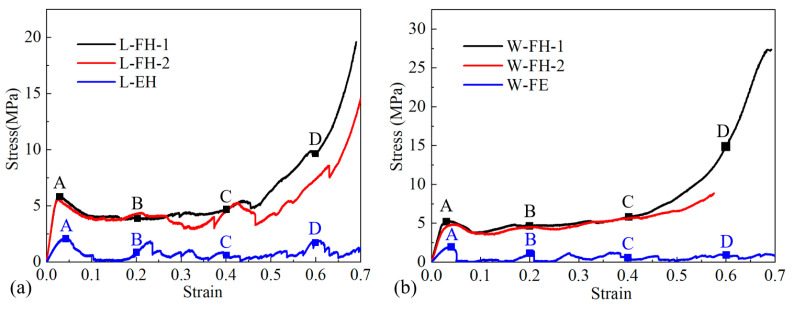
In-plane compressive stress–strain of absorbent-foam-filled honeycomb and empty honeycomb. (**a**) Nominal stress–strain curves of specimens along L-direction; (**b**) nominal stress–strain curves of specimens along W-direction.

**Figure 5 polymers-12-02059-f005:**
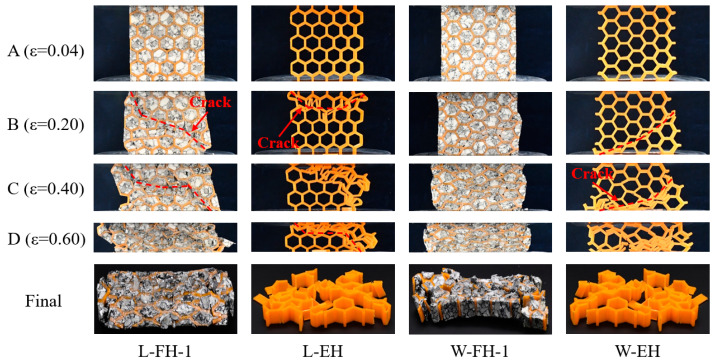
In-plane compressive typical deformation histories of absorbent-foam-filled honeycomb and empty honeycomb, under L- and W-direction, at the selected points marked in Figure 5.

**Figure 6 polymers-12-02059-f006:**
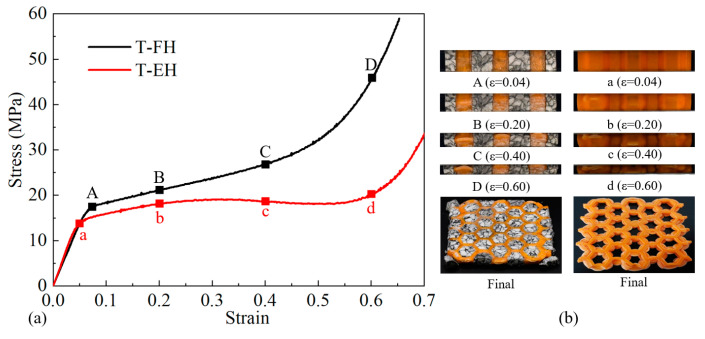
Results of out-of-plane compression test. (**a**) Out-of-plane compressive stress–strain curves of specimens. (**b**) Corresponded deformation histories of specimens.

**Figure 7 polymers-12-02059-f007:**
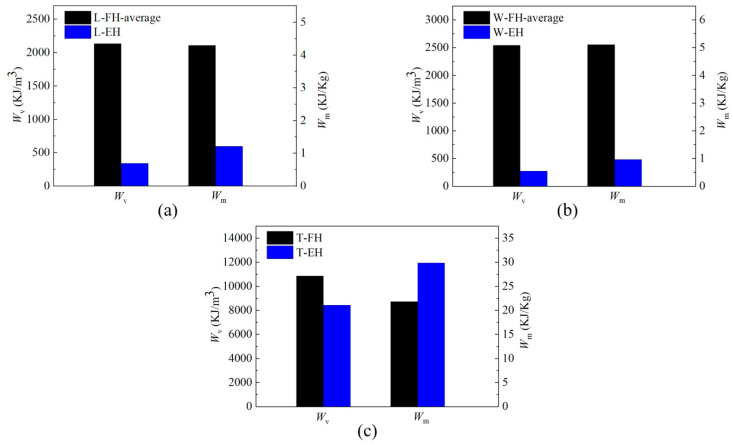
Comparison of energy absorption per unit volume ($W_{v}$) and per unit mass ($W_{m}$ ) of specimens in three directions: (**a**) L-direction, (**b**) W-direction, and (**c**) T-direction.

**Figure 8 polymers-12-02059-f008:**
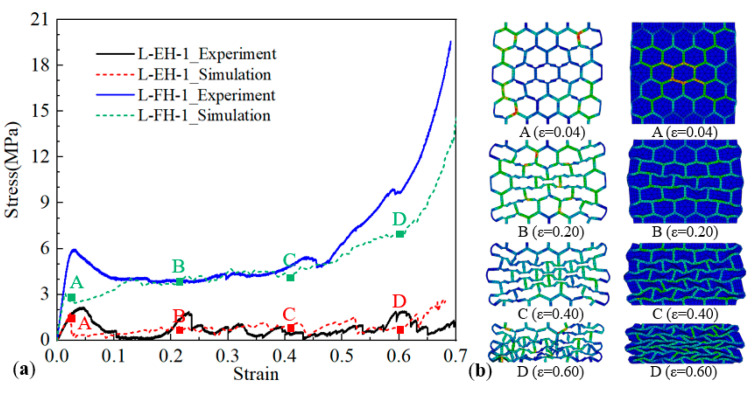
FE-simulated results compared with the experimental results along L-direction: (**a**) comparison of experimental and simulated compressive stress–strain curves, and (**b**) corresponded deformation process at the selected points.

**Figure 9 polymers-12-02059-f009:**
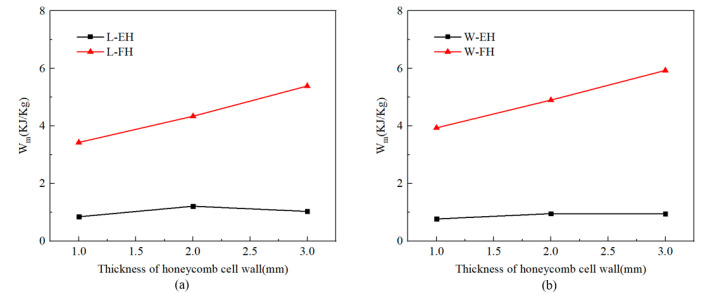
Comparison of energy-absorption characteristics with different honeycomb cell-wall thicknesses: (**a**) L-direction and (**b**) W-direction.

**Table 1 polymers-12-02059-t001:** Summary of density ($\rho_{c}$), elastic modulus (E), compressive strength ($\sigma_{peak}$), energy absorption per unit volume ($W_{v}$), and per unit mass ($W_{m}$) of the specimens.

Specimen	$\rho_{c}(Kg/m^{3})$	E	$\sigma_{peak}(MPa)$	$W_{v}(KJ/m^{3})$	$W_{m}(KJ/Kg)$
L-FH-1	496.97	283.47	5.96	2132	4.29
L-FH-2	495.01	285.44	5.65	1998	4.04
L-EH	281.82	71.77	2.16	341	1.21
W-FH-1	497.46	238.06	5.60	2542	5.11
W-FH-2	496.46	224.74	5.65	2351	4.74
W-EH	283.33	74.25	2.01	272	0.96
T-FH	497.34	276.94	17.35		21.82
T-EH	282.90	309.26	14.05	8439	29.83

Notes: L, W, or T means under L-, W-, and T-direction respectively; EH or FH means empty honeycomb or foam-filled honeycomb, respectively; 1 or 2 means parallel specimens. All specimens have same dimension parameter (81.87 mm × 78.78 mm × 15.51 mm).

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
