# Peer review of "Effect of Absorbent Foam Filling on Mechanical Behaviors of 3D-Printed Honeycombs"

_polymers, 2020, doi:10.3390/polym12092059_

Round 1

Reviewer 1 Report

The idea of the paper “Effect of absorbent foam filling on mechanical  behaviors of 3D printed honeycombs” is to use the same skeleton for supporting the structure mechanically and provide wanted porosity to the foam like structure to suppress reflection.

It make sense to think whether the deviation from the ideal hexagonal shape (typical for 3d printing architectures)  will influence the mechanical and electromagnetic properties of the structure.

Is it possible to optimize geometry in the way that reflection will be suppressed in wider frequency range?

Please compare the absorption ability of this structure in view of its density and thickness with carbon foams and graphene based heterostructures respectively (see e.g. CARBON, 122 (2017) 217-227 https://doi.org/10.1016/j.carbon.2017.06.080 for foams and Appl.Phys. Lett. 108, 123101 (2016) http://dx .doi.org/10.1063/ 1.4944531  for graphene.

Reviewer 2 Report

This manuscript describes a 3D printed PLA honeycomb architecture filled with an absorbent PMI foam. The mechanical behavior was modeled and tested empirically. Additionally, some data on electromagnetic radiation absorption is provided.

Overall, the manuscript requires substantial revision in all areas before an adequate review can be completed.

  • The motivation behind this work is lacking. What are the objectives and design requirements? How would this be used? How can you tell if it is useful or better than existing structures?
  • Experimental details are nearly completely absent. What kind of 3D printing technology was used? Details describing measurement and analysis of electromagnetic wave absorption needs to be added. Figures should be improved to have consistent resolution, font sizes, axes, etc.
  • In general, the organization of the paper is difficult to follow.
  • There is no description of the coupling between the foam and the PLA.
  • What is the novelty here?

Reviewer 3 Report

Title ia appropriate and reflects the essence of the paper.

The Abstract is sufficient and informative

Introduction

Please avoid bulk citation line 35 [1-7] please use no more than 2 references for one fact/statement. References 1-7 is appropriate, but please add more details/facts about it.

Please add some motivation for the Electromagnetic absorber motivation design.

Experiments better rename to the “Materials and methods”

Figure 2 for the reader's convenience you could add the scale bar.

Electromagnetic Wave Absorption – references  8 and is good, but 36 provide nothing information regarding the composition of the absorber.

but still add the used equipment description and key parameters (band range, t.c.).

Fig 2 – is taken 1:1 from paper [36]  written permission should be for using.

The electromagnetic properties in the paper not discussed at all . Probably this image should be omitted.

Please add information about absorber

Part 3. Numerical Simulations -  should be 2.4. as part of Materials and methods.

Lines 155 and 159 – what is difference between L-FH-1 – Why defined twice for the same abbreviation?

Please USE ONCE CLEAR designation for all used specimens type BEFORE firs use ( in table 1). Please do not repeat full name each time. What is the L-FH-2 and W-FH-2  specimens ??? (just parallel specimens?)

Figure 9 please add explanation or the L-EH, L-EH-FE, L-FH an L-EH-FE  abbreviation in the figure caption.

It would be interesting to see on one or two graphs combined together compression curves for measured and calculated results, to get some correlation between it.

Actually discussion part is completely missing.

Round 2

Reviewer 2 Report

This manuscript has been improved, but there are still details and points lacking.

  • How was the foam cut? How do you ensure uniformity?
  • Is the foam compressed within the 3D printed honeycomb structure? Is there space between the edge of the foam and the honeycomb structure? How much pressure is the foam exerting on the walls of the honeycomb structure? How would this influence your results?
  • Please provide additional parameters for the 3D printing process. 

Author Response

Please see as attached!

Reviewer 3 Report

I see very minor drawbacks  -  mainly punctuation and spacing:

1.Lines 93-95 should be NO space after brackets  "(" and not before ")"

Should be not "( Hc = 15.51mm )" BUT "(Hc = 15.51mm)" 

is necessary to check WHOLE the manuscript for this mistake (Line 241, 251-256

2.Incorrect bibliography design: N. Surname 

but [3] Riccio, A. Raimondo, S. Saputo, A. Sellitto, M. Battaglia, G. Petrone.

Please check all bibliography

3. all images are blurred (maybe it is only in the before-publishing version but PLEASE CHECK !)

Author Response

Please see as attached.
